# Relationship of Desmoplastic Reaction and Tumour Budding in Primary and Lung Metastatic Lesions of Colorectal Cancer and Their Prognostic Significance

**DOI:** 10.3390/cancers17040583

**Published:** 2025-02-08

**Authors:** Toshinori Kobayashi, Mitsuaki Ishida, Hiroshi Matsui, Hiroki Uehara, Shoichiro I, Norikazu Yamada, Yuto Igarashi, Chie Hagiwara, Yoshihiro Mori, Yohei Taniguchi, Tomohito Saito, Haruaki Hino, Yoshinobu Hirose, Tomohiro Murakawa, Jun Watanabe

**Affiliations:** 1Department of Colorectal Surgery, Kansai Medical University, 2-5-1, Shinmachi, Hirakata 573-1010, Japan; ueharah@hirakata.kmu.ac.jp (H.U.); ishoi@hirakata.kmu.ac.jp (S.I.); yamadan@hirakata.kmu.ac.jp (N.Y.); igarashy@hirakata.kmu.ac.jp (Y.I.); hagiwach@hirakata.kmu.ac.jp (C.H.); moriyo@hirakata.kmu.ac.jp (Y.M.); watanabj@hirakata.kmu.ac.jp (J.W.); 2Department of Pathology, Osaka Medical and Pharmaceutical University, 2-7, Daigaku-machi, Takatsuki 569-8686, Japan; yoshinobu.hirose@ompu.ac.jp; 3Department of Thoracic Surgery, Kansai Medical University, 2-5-1, Shinmachi, Hirakata 573-1010, Japan; mathiros@hirakata.kmu.ac.jp (H.M.); taniguyo@hirakata.kmu.ac.jp (Y.T.); saitotom@hirakata.kmu.ac.jp (T.S.); hinoh@hirakata.kmu.ac.jp (H.H.); murakawt@hirakata.kmu.ac.jp (T.M.)

**Keywords:** colorectal cancer, lung metastasis, desmoplastic reaction, tumour budding, prognosis, overall survival

## Abstract

Colorectal cancer (CRC) is a significant global health burden, with substantial morbidity and mortality rates. Hence, there is a significant need for prognostic measures to enhance the care and management of patients. However, no study has examined the relationship between histological indicators, including desmoplastic reaction (DR) and tumour budding (TB), in metastatic lung lesions of CRC. Findings from this study demonstrated that higher TB and immature-type DR in metastatic lung lesions were significantly poor prognostic indicators. Accordingly, histological indicators of metastatic lung lesions in CRC may provide important prognostic information for oncologists that can help promote patient care, management, and overall survival of patients with CRC. Additional studies are required to elucidate the mechanisms underlying DR in metastatic lung lesions.

## 1. Introduction

Colorectal cancer (CRC) is a significant global health burden [1]. Despite advancements in treatment modalities, metastasis to distant organs such as the lungs remains a major challenge that significantly affects patient outcomes [2,3,4,5]. Metastasis is a critical event in CRC progression that significantly affects patient prognosis and treatment outcomes. The tumour microenvironment (TME) plays a pivotal role in facilitating the proliferation of metastatic dissemination [6,7]. The TME comprises various cellular and noncellular components that surround cancer cells. This dynamic milieu includes immune cells, fibroblasts, blood vessels, extracellular matrix (ECM), and signalling molecules. Interactions within the TME profoundly influence cancer progression, metastasis, and response to therapy [8,9,10]. Desmoplastic reaction (DR) refers to the stromal response characterised by the deposition of various proportions of ECM components, particularly collagen, fibroblasts, and other stromal cells around the carcinoma cells [11]. The prognostic significance of DR has been reported in CRC for the first time, highlighting DR type as a strong prognostic indicator for patients with pT3 or pT4 CRC [12,13], as well as other carcinoma types [14,15,16]. The type of DR is believed to reflect the TME and influence prognosis [12]. DR is classified into immature (IM), intermediate, and mature types, and it has been established that IM-type DR was significantly correlated with a poor prognosis in CRC [12].

The TME is also formed in metastatic lesions, and DR is observed in metastatic sites such as the lymph nodes and liver [17,18]. Previous studies have demonstrated a similarity in the TME between the primary tumour and metastatic sites of CRC [17,18]. Ao et al. reported that the types of DR of the primary lesions were significantly correlated with those of lymph node and liver metastasis, and IM-type DR in the lymph node or liver metastatic sites showed worse relapse-free survival compared to other DR types [18]. Although the lungs are one of the most frequent metastatic sites of CRC, no study has examined the relationship between DR in primary and metastatic lung lesions and the prognostic significance of DR in metastatic lung lesions of CRC.

Tumour budding (TB), defined as one to four carcinoma cell clusters separated from the main tumour, is associated with invasiveness and metastasis [19]. TB of primary CRC lesions is a prognostic indicator [19]. Karjula et al. reported that TB in metastatic lung lesions had no prognostic significance and was not significantly correlated with that in primary lesions [20]. It has been reported that the degree of TB correlates with the type of DR in primary CRC [18]. However, no study has examined the relationship between DR and TB in metastatic lung lesions of CRC.

The purpose of the present study was to investigate the relationship between DR and TB and their prognostic significance in primary and metastatic lung lesions in patients with CRC. The results of the present study can provide important information for treatment in patients with lung metastasis of CRC.

## 2. Materials and Methods

### 2.1. Patient Selection

We selected consecutive patients with pT3 or pT4 CRC with lung metastasis who underwent surgical resection of the primary lesions and synchronous or metachronous metastatic lung lesions at the Department of Surgery or Department of Thoracic Surgery at Kansai Medical University Hospital between January 2016 and December 2020. This was because DR is defined in patients with pT3 or pT4 CRC [12,13]. The inclusion criteria for the present study were patients with CRC who underwent surgical resection of the primary and metastatic lung lesions, with the re-evaluation of the histopathological features of both lesions available. The exclusion criteria were patients with CRC who underwent surgical resection of the primary and metastatic lung lesions and in whom re-evaluation of the histopathological features of both lesions was unavailable.

This retrospective study was reviewed and approved by the Institutional Review Board of Kansai Medical University Hospital, Osaka, Japan (approval number: 2021357). This study was conducted in accordance with the principles of the Declaration of Helsinki. The Institutional Review Board of Kansai Medical University Hospital waived the requirement for informed consent because of the retrospective nature of the study. Moreover, this study did not include patients <18 years. Data regarding this study, such as the inclusion criteria and the options to opt out, were provided on the institutional website (https://hp.kmu.ac.jp/upload/clinical_research/document/2671t800001356c-att/a1649211737374.pdf (accessed on 6 February 2025)).

### 2.2. Histopathological Analysis

Surgically resected specimens of both the colorectum and the lungs were fixed in 10% buffered neutral formalin, sectioned, and stained with haematoxylin and eosin. Two researchers (Toshinori Kobayashi and Mitsuaki Ishida) independently evaluated the histopathological features of all tumour slides. The staging of patients with CRC was conducted according to the American Joint Committee on Cancer 8th edition staging system, utilising parameters such as the depth of tumour invasion (pT), tumour lymph node, and distant metastasis (pN and M) [21].

DR was classified into IM, intermediate, and mature types according to previous reports [12,13]. The IM type is histopathologically defined by the presence of myxoid stroma (amorphous mucoid material) at a magnification greater than 400× at the invasive front. The intermediate type is characterised by the presence of keloid-like collagen (thick bundles of hypocellular hyalinized collagen) without myxoid stroma. The mature type is defined by the absence of both myxoid stroma and keloid-like collagen.

TBs were also evaluated using the same histopathological method as previously reported [22]. Tumour buds were defined as the presence of a single tumour cell or a cluster of up to four tumour cells. Tumours with 0–4 buds at the invasive front were classified as TB1, 5–9 buds as TB2, and more than 10 buds as TB3.

Histological differentiation was categorised as well-, moderately, or poorly differentiated. Well-differentiated was characterised by well-formed tubular units with complete and easily discernible borders; moderately differentiated was characterised by incomplete and ill-defined glandular borders, fusion of glands, or irregular multi-lumen cribriform pattern; and poorly differentiated was characterised by non-glandular formation with cord-like, nested, solid, or individual cell infiltration [23]. The least differentiated component was defined as the differentiation grade.

In patients with more than one lung metastasis, the highest or least histological parameters were defined as the histopathological indicators.

### 2.3. Statistical Analyses

All analyses were performed using the JMP software version 17.0.0 (SAS Institute, Cary, NC, USA). The correlations between the two groups were analysed using the chi-square test or Fisher’s exact test for categorical variables, and concordance rates were calculated using Cohen’s kappa (κ) coefficient to assess agreement. Overall survival (OS) rates were compared using the Kaplan–Meier method, and *p*-values were calculated using the log-rank test. Statistical significance was set at *p* < 0.05. Every possible comparison between subgroups was considered.

## 3. Results

### 3.1. Patient Characteristics

The cohort included 40 patients with pT3 or pT4 CRC who underwent surgical resection of synchronous or metachronous lung metastases. Table 1 summarises the clinicopathological features of the study cohort. The present cohort included 21 males (52.5%) and 19 females (47.5%). The median age at the time of surgery for lung metastasis was 70 years (range 47–88 years old). The primary CRC was located on the right side, left side, and rectum in 6, 9, and 25 patients, respectively.

The pT stages of primary CRC featured 31 and 9 patients in pT3 and pT4, respectively. Histological differentiation types of the primary lesions were well- (13 patients, 32.5%) or moderately differentiated (27 patients, 67.5%).

Synchronous lung metastases were noted in 14 patients (35.0%), whereas the remaining 26 patients (65.0%) experienced metachronous lung metastases. Amongst patients who underwent lung metastasis resection, 11 patients (27.5%) received preoperative chemotherapy. In contrast, postoperative adjuvant chemotherapy was administered to four patients (10%) who achieved R0 resection without preoperative treatment. Timing of metastasis (synchronous or metachronous) was not significantly correlated with OS (*p* = 0.19).

#### 3.1.1. DR and TB in the Primary CRC

Thirty-two patients (80.0%) were classified as having IM DR type, and the remaining eight patients (20.0%) as having non-IM (intermediate or mature) DR type (Table 1).

Fifteen (37.5%), eight (20.0%), and seventeen (42.5%) patients were classified as having TB1, TB2, and TB3, respectively (Table 1).

#### 3.1.2. DR and TB in the Lung Metastasis Lesions

Six patients (15.0%) were classified as having IM DR type (Figure 1A), and the remaining thirty-four patients (85.0%) as having non-IM DR type in the metastatic lung lesions (Table 1).

Thirty-one (77.5%), seven (17.5%), and two (5.0%) patients were classified with TB1, TB2, and TB3, respectively (Table 1, Figure 1B).

### 3.2. Relationship Between DR in the Lung Metastatic Lesions and Clinicopathological Parameters

Moderately differentiated histological type, higher TB in the primary lesions, and presence of lymph node metastasis were associated with IM-type DR in lung metastatic lesions (*p* = 0.022, 0.043, and 0.020, respectively) (Table 2).

Moreover, moderately differentiated histological type and higher TB in metastatic lung lesions were significantly correlated with IM-type DR in metastatic lung lesions (*p* = 0.039 and 0.009, respectively) (Table 2).

### 3.3. Relationship of DR and TB in the Primary and Lung Metastatic Lesions

The type of DR (IM vs. non-IM) showed no significant correlation between the primary and metastatic lesions (*κ* = 0.08, *p* = 0.086). All six patients with IM-type DR in the lung metastatic lesions (synchronous and metachronous in three patients, each) showed IM-type DR in the primary lesions, and the remaining 26 patients with IM-type DR in the primary lesions showed non-IM-type DR in the lung metastatic lesions (Table 2).

Regarding TB, a moderate level of agreement between metastatic lung lesions and primary lesions was statistically demonstrated (*κ* = 0.47, *p* = 0.015). Amongst the seven patients with TB2 in metastatic lung lesions, one and six patients were classified as having TB2 and TB3 primary lesions, respectively. In the two patients with TB3 in metastatic lung lesions, the primary lesion was TB1 (Table 3).

IM-type DR and higher TB were significantly correlated in both metastatic lung lesions (*p* = 0.009) (Table 2 and Table 3).

### 3.4. Prognostic Impact on DR and TB

Figure 2 shows the Kaplan–Meier curves of OS according to DR type (IM vs. non-IM). IM-type DR in metastatic lung lesions was significantly correlated with poor OS (*p* = 0.0020) (Figure 2A). The type of DR (IM vs. non-IM) in primary CRC did not significantly correlate with OS (*p* = 0.84) (Figure 2B).

Figure 3 shows the Kaplan–Meier curves of OS according to TB. Higher TB (TB1 vs. TB2/3) in both metastatic lung lesions and primary CRC was significantly correlated with poor OS (*p* = 0.044, = 0.021, respectively) (Figure 3A,B).

## 4. Discussion

In the present study, we demonstrated that IM-type DR and TB2/3 in metastatic lung lesions were significantly correlated with poor OS. Although the types of DR between primary and metastatic lung lesions in patients with CRC were not significantly correlated, TB was significantly correlated. This is the first study to examine the prognostic significance of DR in metastatic lung lesions and the relationship between DR and primary and metastatic lung lesions in CRC.

DR has been recognised as a powerful prognostic indicator in patients with pT3 or pT4 CRC [12,13]. DR is composed of various amounts of ECM, mainly produced by cancer-associated fibroblasts around the carcinoma cells [8,9,10,11]. The types of DR may represent these TMEs, including cancer-associated fibroblasts, and have been shown to be significant prognostic indicators in patients with CRC; that is, the presence of IM-type DR has the worst prognosis, followed by intermediate and mature types [12,13]. The prognostic significance of DR has also been reported in some types of carcinomas, such as oral and oesophageal carcinomas, as well as CRC [14,16], although the detailed mechanism of the types of DR related to prognosis remains unclear [12,13]. Moreover, our group proposed the first prognostic scoring system based on indicators, including DR, in patients with CRC [23]. Thus, DR may become increasingly important in the prognosis and treatment strategy of patients with CRC.

DR can also be present in the metastatic sites of CRC, and some recent studies have shown a relationship between DR at the primary sites, metastatic lymph nodes, and liver lesions in patients with CRC [17,18]. Ao et al. demonstrated that the types of DR in primary lesions were significantly correlated with those of both lymph node and liver metastatic lesions [17,18]. IM-type DR in lymph node metastatic lesions was significantly correlated with a higher lymph node metastatic status and higher TB but not tumour differentiation grade [18]. IM-type DR in liver metastatic lesions was significantly correlated with tumour differentiation grade but not lymph node metastatic status [18]. IM-type DR in both the metastatic lymph nodes and liver were significantly correlated with poor recurrence-free survival [18]. However, the relationship between DR and primary and metastatic lung lesions in CRC has not yet been elucidated. The results of the present study were not consistent with those of previous reports [17,18] because the types of DR between primary and metastatic lung lesions were not significantly correlated. However, IM-type DR in metastatic lung lesions was significantly correlated with poor OS. This result indicates that IM-type DR in metastatic lung lesions is a useful prognostic indicator for patients with lung metastatic CRC. Although the mechanism of the difference in the types of DR between primary and metastatic lung lesions remains unclear, the environment around the carcinoma cells or tissue structure between the liver and the lung (for example, a large amount of the lung is air space) might influence this difference. Further analysis with a larger cohort is required to clarify the mechanisms underlying this difference.

TB represents the epithelial–mesenchymal transition of carcinoma cells and is associated with invasion and metastasis [19,24,25]. A higher TB level has been established as a poor prognostic indicator in patients with CRC [19,23], as well as other types of carcinomas [24,25,26,27,28,29,30,31]. It has been shown that higher TB is associated with IM-type DR in primary CRC lesions and metastatic lymph nodes [13,18], similar to the results of the present study on metastatic lung lesions. The relationship of TB between the primary lesions and liver metastatic lesions in patients with CRC remains controversial because one report showed a significant association [32], but the other reported the contrary result [33]. The present study demonstrated a significant relationship between TB and primary and lung metastatic lesions in patients with CRC, although a previous study showed contradictory results [20]. Moreover, higher TB in metastatic lung lesions was a significantly poor prognostic indicator in the present cohort, although a previous study showed no significant prognostic difference in the degree of TB in metastatic lung lesions of CRC [20]. Therefore, additional studies with larger cohorts are needed to clarify the prognostic significance of TB in metastatic lung lesions in patients with CRC.

The present study has some limitations. First, it included a relatively small number of patients with lung metastatic CRC from a single institution. Thus, selection bias cannot be ruled out. Additional multi-institutional studies with larger cohorts are needed to clarify the prognostic significance of DR and TB in metastatic lung lesions and the relationship between primary and metastatic lung lesions in patients with CRC. Second, we adapted the same criteria for DR and TB in metastatic lung lesions and primary CRC [12,13,19] because previous reports analysing DR and TB in metastatic lymph nodes and liver lesions used the same criteria [17,18]. In the previous study examining the prognostic significance of TB and its relationship between primary CRC and metastatic lung lesions, TB was classified into low (<5 buds) and high (≥5 buds) categories [20]. Additional studies are required to clarify the validity of these criteria for detecting DR and TB in metastatic lung lesions. These studies might lead to establishing a standardised method for analysing the useful histopathological prognostic indicators of metastatic lung lesions in patients with CRC.

## 5. Conclusions

The present study demonstrated that IM-type DR and TB2/3 in lung metastatic lesions were significantly poor prognostic indicators in patients with CRC. The results of the present study might provide important information for risk stratification and treatment strategies in patients with lung metastatic CRC. Additional studies with larger cohorts are needed to clarify the mechanism underlying this difference and its prognostic significance.

## Figures and Tables

**Figure 1 cancers-17-00583-f001:**
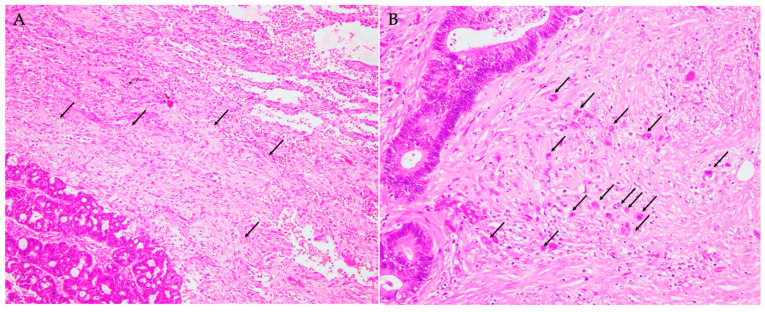
Histopathological features of metastatic colorectal carcinoma in the lung. (**A**) Immature-type desmoplastic reaction. The presence of myxoid stroma (arrows) adjacent to the metastatic colorectal carcinoma cells is noted. Lung tissue is observed in the right upper corner (hematoxylin and eosin staining, 100×). (**B**) Tumour budding. Single carcinoma cells and clusters of up to four carcinoma cells (arrows) are observed around the nests of carcinoma cells formed by large tubular glands (hematoxylin and eosin staining, 200×).

**Figure 2 cancers-17-00583-f002:**
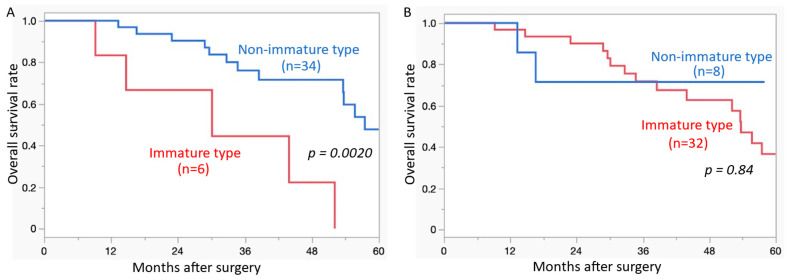
Overall survival curves of colorectal carcinoma with lung metastasis. (**A**) Presence of immature-type desmoplastic reaction in the metastatic lung lesions is a significantly poor prognostic indicator compared to that of the non-immature type. (**B**) Presence of immature-type desmoplastic reaction in the primary colorectal carcinoma lesions is not a significant poor prognostic indicator compared to that of non-immature type.

**Figure 3 cancers-17-00583-f003:**
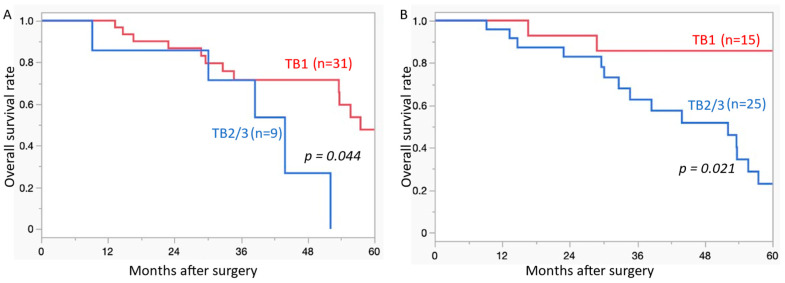
Overall survival curves of colorectal carcinoma with lung metastasis. (**A**) The presence of TB2/3 in the metastatic lung lesions is a significantly poor prognostic indicator compared to that of TB1; (**B**) the presence of TB2/3 in the primary colorectal carcinoma lesions is a significantly poor prognostic indicator compared to that of TB1. TB, tumour budding.

**Table 1 cancers-17-00583-t001:** Clinicopathological features of the study cohort.

			All patients
			n = 40
Age (years)		Median (range)	70 (47–88)
Sex		Male/Female	21/19
Primary tumour			
	Location	Right-sided/Left-sided/Rectum	6/9/25
	Histological type	Well-/Moderately/Poorly differentiated	13/27/0
	T stage	3/4	31/9
	N stage	0/1/2	17/15/8
	Tumour budding	TB 1/2/3	15/8/17
	Desmoplastic reaction	Immature/Non-Immature	32/8
Lung metastasis			
	Timing of metastasis	Synchronous/Metachronous	14/26
	Histological type	Well-/Moderately/Poorly differentiated	11/29/0
	Tumour budding	TB 1/2/3	31/7/2
	Desmoplastic reaction	Immature/Non-Immature	6/34

**Table 2 cancers-17-00583-t002:** Relationship of desmoplastic reactions between primary colorectal cancer and metastatic lung lesions.

Desmoplastic Reaction of Lung Metastasis					
			Non-Immature	Immature	*p* Value
n			34	6	
Age (years)		Median (range)	69.5 (47–88)	72 (63–83)	0.50
Sex		Male/Female	19/15	2/4	0.30
Lung metastasis					
	Timing of metastasis	Synchronous/Metachronous	11/23	3/3	0.40
	Histological type	Well-/Moderately Differentiated	11/23	0 / 6	0.039
	Tumour budding	TB 1/2/3	29/3/2	2/4/0	0.009
Primary tumour					
	Location	Right-sided/Left-sided/Rectum	5/7/22	1/2/3	0.76
	Histological type	Well-/Moderately Differentiated	13/21	0/6	0.022
	T stage	¾	28/6	3/3	0.10
	N stage	0/1/2	16/14/4	1/1/4	0.020
	Tumour budding	TB 1/2/3	15/6/13	0/2/4	0.043
	Desmoplastic reaction	Non-immature/Immature	8/26	0/6	0.086
	Cohen’s kappa (κ) for desmoplastic reaction	0.08 (slight agreement)			

**Table 3 cancers-17-00583-t003:** Relationship of tumour budding between primary colorectal cancer and lung metastatic lesions.

Tumour Budding of Lung Metastasis						
			BD1	BD2	BD3	*p* Value
n			31	7	2	
Age (years)		Median (range)	68 (47–87)	73.5 (58–83)	68	0.79
Sex		Male/Female	14/17	4/3	1/1	0.84
Lung metastasis						
	Timing of metastasis	Synchronous/Metachronous	10/21	3/4	1/1	0.78
	Histological type	Well-/Moderately Differentiated	11/20	0/7	0/2	0.034
	Desmoplastic reaction	Non-immature/Immature	29/2	3/4	2/0	0.009
Primary tumour						
	Location	Right-sided/Left-sided/Rectum	5/4/22	1/4/2	0/1/1	0.12
	Histological type	Well-/Moderately Differentiated	11/20	0/7	2/0	0.0063
	T stage	3/4	25/6	4/3	2/0	0.26
	N stage	0/1/2	13/14/4	2/1/4	2/0/0	0.046
	Desmoplastic reaction	Non-immature/Immature	6/25	1/6	1/1	0.58
	Tumour budding	BD 1/2/3	13/7/11	0/1/6	2/0/0	0.015
	Cohen’s kappa (κ) for tumour budding	0.47 (moderate agreement)				

## Data Availability

Due to the nature of this research, participants in this study could not be contacted about whether the findings could be shared publicly. Thus, supporting data are not available. The datasets generated and analysed during the current study are not publicly available due to the nature of the research but are available from the corresponding author on reasonable request.

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
