# Peer review of "Relationship of Desmoplastic Reaction and Tumour Budding in Primary and Lung Metastatic Lesions of Colorectal Cancer and Their Prognostic Significance"

_cancers, 2025, doi:10.3390/cancers17040583_

Round 1

Reviewer 1 Report

Comments and Suggestions for Authors

Accept in present form

Author Response

Accept in present form

Response:

Thank you for reviewing our manuscript.

No change has been made to the manuscript.

Reviewer 2 Report

Comments and Suggestions for Authors

The article entitled “Relationship of desmoplastic reaction and tumour budding between the primary and lung metastatic lesions of colorectal cancer and their prognostic significance” describes the observations of a just morphological/histological study in relation to the development of metastasis and of the overall survival time of a rather low number of patients (N = 40) with colorectal cancer (CRC). The reported relationships are based on well conducted statistical analyses. Otherwise, no mention is made to genetic and/or molecular characteristics.

Assuming that the loss of cellular differentiation of the carcinoma cells and the accompanying desmoplastic reaction (DR) and that tumour budding (TB) are related to tumour aggressiveness, the conclusion that “IM-type DR and TB2/3 in lung metastatic lesions were significantly poor prognostic indicators in patients with CRC” could be envisaged as anticipated and taken for granted. But true science is based on data, and this wotk is based on observational data. These data may be of help for oncologists for better patient care and management.

 In spite of the limitations of this work, as acknowledged by the authors and stated in the paper, and as a starting point for further research, I consider that it is worth of publishing.

Author Response

Response to comments of Reviewer 2:

The article entitled “Relationship of desmoplastic reaction and tumour budding between the primary and lung metastatic lesions of colorectal cancer and their prognostic significance” describes the observations of a just morphological/histological study in relation to the development of metastasis and of the overall survival time of a rather low number of patients (N = 40) with colorectal cancer (CRC). The reported relationships are based on well conducted statistical analyses. Otherwise, no mention is made to genetic and/or molecular characteristics.

Assuming that the loss of cellular differentiation of the carcinoma cells and the accompanying desmoplastic reaction (DR) and that tumour budding (TB) are related to tumour aggressiveness, the conclusion that “IM-type DR and TB2/3 in lung metastatic lesions were significantly poor prognostic indicators in patients with CRC” could be envisaged as anticipated and taken for granted. But true science is based on data, and this work is based on observational data. These data may be of help for oncologists for better patient care and management.

 In spite of the limitations of this work, as acknowledged by the authors and stated in the paper, and as a starting point for further research, I consider that it is worth of publishing.

Response:

Thank you for reviewing our manuscript. We acknowledge your kind comments.

No change has been made to the manuscript.

Reviewer 3 Report

Comments and Suggestions for Authors

Paper titled (Relationship of desmoplastic reaction and tumour budding between the primary and lung metastatic lesions of colorectal cancer and their prognostic significance) by Toshinori Kobayashi et al. studied colorectal cancer and explored the relationship between domestic reaction and tumor budding & the occurence of primary and lung metastatic lesions and its prognostic value. authors used histological indicators and cloncluded that Histological indicators of metastatic lung lesions in CRC may provide important prognostic information. This is a novel and important study and the data are supoorting to the conclusion.

1- In abstract: please add some numerical values from key findings of this study

2- keywords: may add (prognosis)

3- Intorduction: first paragraph at line 49 starts with (Colorectal cancer (CRC) is a significant global health burden, with substantial morbid- 49 ity and mortality rates. ) till its end 

is long and did not introduce tumore metastasis well 

4- Mention clearly the inclusion and exclusion criteria for this study

5- Write the limitations and future prespectives for this study 

6- Write the clinical potential of this study 

7- The aim of the study at the end of the introduction should be rewritten, explore the novelty & mention how authors achieved this aim.

8-  Use appropriate abbreviations for minutes, seconds...etc

9- Every abbreviation in figures should be explained in the figure legend to be self explanatory & stands alone.
10- Authors should confirm in methods that "every possible comparison between the study groups was considered" and apply this in results
11- Mention "n" in each illustration individually

Author Response

Response to comments of Reviewer 3:

Thank you for reviewing our manuscript. We appreciate your constructive comments, which were very helpful in revising our manuscript. We have revised the manuscript following the concerns highlighted by you. 

Paper titled (Relationship of desmoplastic reaction and tumour budding between the primary and lung metastatic lesions of colorectal cancer and their prognostic significance) by Toshinori Kobayashi et al. studied colorectal cancer and explored the relationship between domestic reaction and tumor budding & the occurence of primary and lung metastatic lesions and its prognostic value. authors used histological indicators and concluded that Histological indicators of metastatic lung lesions in CRC may provide important prognostic information. This is a novel and important study and the data are supporting to the conclusion.

1- In abstract: please add some numerical values from key findings of this study

Response:

As you suggested, we added some numerical values of the present study in the Abstract.

No change was made in numerical values showing significance differences in prognosis (P values) in the Abstract.

“Results: Overall, 40 patients with CRC (males, 21; females, 19; median age 70 years; right-side colon 6; left-side colon 9; rectum 25; and pT3, 31; pT4, 9) were evaluated.”

(page 1, lines 37-38)

2- keywords: may add (prognosis)

Response:

As you suggested, we added “prognosis” to the keywords.

Keywords: colorectal cancer; lung metastasis; desmoplastic reaction; tumour budding; prognosis; overall survival.” (page 2, line 46)

3- Introduction: first paragraph at line 49 starts with (Colorectal cancer (CRC) is a significant global health burden, with substantial morbidity and mortality rates. ) till its end is long and did not introduce tumor metastasis well 

Response:

As you suggested, we revised and shortened the sentence in the first paragraph of the Introduction.

“Colorectal cancer (CRC) is a significant global health burden [1]. Despite advancements in treatment modalities, metastasis to distant organs such as the lungs remains a major challenge that significantly affects patient outcomes [2–5].” (page 2, lines 50-52)

4- Mention clearly the inclusion and exclusion criteria for this study

Response:

As you mentioned, we added the comments regarding the inclusion and exclusion criteria of the present study.

“The inclusion criteria for the present study were patients with CRC who underwent surgical resection of the primary and metastatic lung lesions with the re-evaluation of the histopathological features of both lesions available. The exclusion criteria were patients with CRC who underwent surgical resection of the primary and metastatic lung lesions and in whom re-evaluation of the histopathological features of both lesions was unavailable.” (page 3, lines 93-98)

5- Write the limitations and future perspectives for this study 

Response:

As you suggested, we added the comment regarding the limitations and further perspective of the present study in the Discussion section.

“Additional multi-institutional studies with larger cohorts are needed to clarify the prognostic significance of DR and TB in the metastatic lung lesions and the relationship between primary and metastatic lung lesions in patients with CRC.” (page 10)

“These studies might lead to establishing a standardised method for analysing the useful histopathological prognostic indicators of metastatic lung lesions in patients with CRC.” (page 10, lines 296-298, and lines 305-207)

6- Write the clinical potential of this study 

Response:

As you mentioned, we added a comment regarding the clinical potential of the present study in the Conclusion section.

“The results of the present study might provide important information for risk stratification and treatment strategies in patients with lung metastatic CRC.” (page 11, lines 311-313)

7- The aim of the study at the end of the introduction should be rewritten, explore the novelty & mention how authors achieved this aim.

Response:

As you mentioned, we rewrote the last paragraph of the Introduction, describing the novelty and mention of the present study.

“The purpose of the present study was to investigate the association between DR and TB and their prognostic significance in primary and metastatic lung lesions in patients with CRC. The results of the present study might provide important information for treatment in patients with lung metastasis of CRC.” (pages 3-4, lines 83-86)

8-  Use appropriate abbreviations for minutes, seconds...etc

Response:

As you suggested, we revised the abbreviations in the manuscript.

“The types of DR may represent these TME, including cancer-associated fibroblasts, and have been shown to be significant prognostic indicators in patients with CRC; that is, the presence of IM-type DR has the worst prognosis, followed by intermediate and mature types.” (page 9, 248-251)

9- Every abbreviation in figures should be explained in the figure legend to be self explanatory & stands alone.

Response:

As you mentioned, we have described the abbreviations in the figure legends.

“TB, tumour budding.” (Figure 3, page 9)

10- Authors should confirm in methods that "every possible comparison between the study groups was considered" and apply this in results

Response:

As you suggested, we added the comment in the Materials and Methods, and also added the results.

“Every possible comparison between subgroups was considered.” (2.3. Statistical analysis, page 4, 143-144)

“Timing of metastasis (synchronous or metachronous) was not significantly correlated with OS (P=0.19, data not shown). (3.1. Patient characteristics, page 5, lines 164-166)

11- Mention "n" in each illustration individually

Response:

As you suggested, we added “n” in Figures 2 and 3.

Round 2

Reviewer 3 Report

Comments and Suggestions for Authors

The revised version of paper titled (Relationship of desmoplastic reaction and tumour budding between the primary and lung metastatic lesions of colorectal cancer and their prognostic significance by Doctor Toshinori Kobayashi  and his colleagues was very well revised and improved compared to the original submission. I wish that the authors take care of these points in their future work.

Now I can recommed this nice article for publication in Cancers.